

# Deep learning-based automatic action extraction from structured chemical synthesis procedures

Mantas Vaškevičius[1,2], Jurgita Kapočiūtė-Dzikienė[1], Arnas Vaškevičius[3] and Liudas Šlepikas[2]

[1] Department of Applied Informatics, Vytautas Magnus University, Kaunas, Lithuania
[2] JSC Synhet, Kaunas, Lithuania
[3] Faculty of Mechanical Engineering and Design, Kaunas University of Technology, Kaunas, Lithuania

## ABSTRACT

This article proposes a methodology that uses machine learning algorithms to extract actions from structured chemical synthesis procedures, thereby bridging the gap between chemistry and natural language processing. The proposed pipeline combines ML algorithms and scripts to extract relevant data from USPTO and EPO patents, which helps transform experimental procedures into structured actions. This pipeline includes two primary tasks: classifying patent paragraphs to select chemical procedures and converting chemical procedure sentences into a structured, simplified format. We employ artificial neural networks such as long short-term memory, bidirectional LSTMs, transformers, and fine-tuned T5. Our results show that the bidirectional LSTM classifier achieved the highest accuracy of 0.939 in the first task, while the Transformer model attained the highest BLEU score of 0.951 in the second task. The developed pipeline enables the creation of a dataset of chemical reactions and their procedures in a structured format, facilitating the application of AI-based approaches to streamline synthetic pathways, predict reaction outcomes, and optimize experimental conditions. Furthermore, the developed pipeline allows for creating a structured dataset of chemical reactions and procedures, making it easier for researchers to access and utilize the valuable information in synthesis procedures.

## INTRODUCTION

Organic chemistry constitutes a crucial discipline within the scientific domain, significantly influencing numerous aspects of human existence. As a result of its numerous breakthroughs, this field has facilitated the synthesis of vital pharmaceuticals, energy sources, and materials that underpin contemporary life. Instances of its contributions range from the inception of antibiotics to the generation of synthetic polymers, thereby underscoring the indispensable role of organic chemistry in propelling societal progress. In recent years, the field of chemistry has witnessed an increasing integration with artificial

Corresponding author
Mantas Vaškevičius,
mantas.vaskevicius@vdu.lt

intelligence (AI) and machine learning (ML) technologies, leading to the development of novel tools and methodologies that harness the capabilities of AI and ML to expedite and enhance chemical research (*Segler, Preuss & Waller, 2018*; *Gómez-Bombarelli et al., 2018*). One notable application of this interdisciplinary approach is predicting chemical properties and reactivity. By training ML algorithms on datasets of chemical structures and their associated properties, researchers can predict the properties of novel molecules, thereby conserving time and resources in the laboratory (*Rupp et al., 2012*). Chemistry and AI have also been synergistically employed in optimizing synthetic pathways. ML algorithms can analyze large datasets of chemical reactions, subsequently identifying patterns and trends that facilitate the design of innovative synthesis pathways (*Szymkuć et al., 2016*). Such models let us discover efficient and cost-effective routes to produce target molecules, a crucial factor in developing new drugs and materials (*Schwaller et al., 2018*). By capitalizing on ML algorithms, a deeper understanding of chemical reactions can be attained, ultimately improving the efficiency and cost-effectiveness of chemical synthesis.

Cheminformatics has emerged as an integral aspect of modern chemical research, relying heavily on extensive datasets necessary to train ML algorithms, which can identify patterns and trends in data that humans may not discern. In addition, large datasets are needed to accurately represent the diversity of chemical structures worldwide, as millions of known chemical compounds possess unique physical and chemical properties. Comprehensive datasets offer an in-depth understanding of chemical phenomena, as they facilitate the identification of patterns and trends that may not be apparent in less diverse datasets, which lack coverage of a wide array of reactions. Our research mainly focuses on synthesis procedures that provide a detailed description of the steps and conditions that are required to carry out a chemical reaction or synthesis. These instructions serve as resources for chemists seeking to replicate and expand upon experiments and AI and data scientists aiming to extract information about reactants, products, synthesis conditions. By harnessing the wealth of data in synthesis procedures, researchers can employ AI-based approaches to streamline synthetic pathways, predict reaction outcomes, and optimize experimental conditions, ultimately accelerating the pace of discovery in chemistry and related fields.

This research aims to develop a methodology for extracting actions from structured chemical synthesis procedures. The methodology includes a pipeline that combines machine ML algorithms and scripts to extract relevant data from the United States Patent and Trademark Office (USPTO) (https://www.uspto.gov, accessed 1 of September 2022) and European Patent Office (https://www.epo.org, accessed on 3 of September 2022) patents by collecting, processing, and transforming experimental procedures into a series of structured actions. The pipeline has two primary tasks solved by ML algorithms: (1) classification of patent paragraphs to select chemical procedures and (2) sequence-to-sequence (seq2seq) generation of chemical procedure sentences into a structured, simplified format. The designed pipeline enables the use of raw patent data to create a dataset of chemical reactions and their procedures in a structured format, allowing for easy modification or incorporation of new patents. This article describes the methods and

algorithms used in ML-based tasks, such as vectorization, model fine-tuning, training, and evaluation. Task 1 involves classifying input data, such as paragraphs of text, into chemical synthesis procedures or other patent-related information. For this classification problem, various types of artificial neural networks (ANNs) were experimentally investigated, including gated recurrent units (GRU), long short-term memory (LSTM), bidirectional LSTM (BiLSTM), and transformers. Task 2 focuses on converting sequential input data, such as sentences of a procedure, into sequential output data consisting of structured procedures with only actions and action parameters. Solving this seq2seq problem required experimental investigation of such ANNs-based methodologies as LSTM, BiLSTM, transformers, and T5 models. We explored various architectures and methodologies for information extraction resulting in detecting the optimal ones, and examined how researchers could effectively employ this pipeline to extract synthesis-related information from patent documents.

## RELATED WORK

This section reviews the literature related to our solving problem by focusing on these topics: extraction of information from chemistry patents, usage of chemical data from patents, text vectorization, and models for language tasks.

Most research about extracting chemical information from patents is based on D. M. Lowe's work. He has significantly contributed to cheminformatics by developing an open-source toolkit for extracting chemical information from scientific literature, including patents. The tool utilizes a combination of ML algorithms, natural language processing, and rule-based techniques to identify and extract chemical entities, reactions, and other relevant information from text and tables. In 2017 an open-source collection of chemical reactions extracted from the USPTO patents issued from 1976–2016 was published and contained compounds and actions extracted from synthesis procedures (*Lowe, 2017*). The software has been adapted and elaborated by the NextMove Software company, which offers a commercial dataset, *Pistachio*, containing millions of chemical reactions that are used for reaction prediction, synthesis planning, and other cheminformatics tasks (https://www.nextmovesoftware.com/pistachio.html, accessed 9 of June 2022). Also, a highly curated commercial tool, *Reaxys*, is available for many similar applications (*Goodman, 2009*). A Web-only chemical tool for retrieval is *ChemSpider SyntheticPages* (*Pence & Williams, 2010*), which offers access to a large database of synthetic chemistry.

Much of the available synthesis procedure data comes from patents because patent applications often include detailed experimental procedures. This level of detail is crucial for computational chemistry and cheminformatics, as it allows them to develop accurate rule-based methods or train ML models for information extraction. The volume of patents and their diverse sources make patent databases rich in synthesis procedures that can be leveraged for computational chemistry and cheminformatics research. The development of *ChemPU*, a tool for organizing structured reaction datasets, has been instrumental in this regard (*Hammer et al., 2021*). Datasets and such tools provide opportunities for chemical synthesis and exploration of the chemical space with automated synthesis robotics and

instruments (*Rohrbach et al., 2022*). Researchers have progressed in various types of chemical information extraction from patents in recent years. For example, a novel patent information extraction framework uses two deep-learning models for entity identification and semantic relation extraction (*Chen et al., 2020*). Another article describes an automated system that extracts chemical entities from patents and classifies their relevance, with high performance in compound recognition and relevancy classification. The study highlights the importance of considering the relevancy of compounds in a patent's context and the potential of automatic text-mining approaches to extract and annotate relevant chemical compounds (*Akhondi et al., 2019*). A deep-learning method has been developed by IBM for the conversion of unstructured experimental procedures in English into structured action sequences for chemical reactions using a transformer-based seq2seq model. The model is pre-trained on large-scale data generated with a rule-based natural language processing approach, yielding high accuracy in predicting action sequences (*Vaucher et al., 2020*). A specialized tool, *ChemU*, has been developed for identifying and extracting chemical entities and related information from unstructured text, such as scientific literature and patents. It employs advanced natural language processing techniques, including ML algorithms and pre-trained language models, to accurately recognize chemical names, formulas, and properties (*He et al., 2020*; *He et al., 2021*; *He et al., 2021*). There has been a significant amount of research on synthesis procedures and patent extraction in natural language processing and AI. The studies discussed above highlight the potential of ML algorithms for extracting relevant information from patent databases, which can have important implications for chemistry and beyond.

Researchers working on computational chemistry have utilized information from synthesis procedure datasets to predict the outcomes of chemical reactions, design new compounds with desired properties, and optimize reaction conditions. ML techniques, such as ANNs and decision trees, have been employed to analyze the wealth of data available in these datasets, enabling the development of predictive models for various chemical processes (*Keith et al., 2021*). For example, *Segler, Preuss & Waller (2018)* used a Monte Carlo tree search algorithm and a deep neural network to perform retrosynthetic analysis and suggest synthetic routes for target molecules. Furthermore, studies have explored the optimization of reaction conditions using data-driven approaches, such as the work by *Ahneman et al. (2018)* where they applied ML to predict the optimal conditions for photocatalytic reactions. The USPTO dataset was used to train the Molecular Transformer, a seq2seq model based on the Transformer architecture, to predict chemical reaction outcomes. The model demonstrated high accuracy and robustness in predicting reaction products, even in the presence of uncertainty (*Schwaller et al., 2019*). Another study used the USPTO dataset to train ML models to predict reaction outcomes. The authors compared various algorithms, including feed-forward neural networks, random forests, and graph convolutional neural networks (*Coley et al., 2017*). A natural language processing architecture was proposed to predict reaction yields using an encoder

transformer model combined with a regression layer and demonstrated outstanding prediction performance on two high-throughput experiment reaction sets (*Schwaller et al., 2021*). A new approach called RetroTRAE for predicting efficient synthetic routes for a target molecule was presented and yields a top-1 accuracy of 58.3% on the USPTO test dataset and outperforms other state-of-the-art neural machine translation-based methods (*Ucak et al., 2022*). Another article proposes a deep learning model that combines a popular cheminformatics reaction representation called the condensed graph of reaction (CGR) with a recent graph convolutional neural network (GCNN) architecture to estimate chemical reaction properties. The approach outperforms current state-of-the-art models in accuracy, applies even to imbalanced reactions, and possesses excellent predictive capabilities for diverse target properties, and curated benchmark data sets are made available online, free of charge, and open source (*Heid & Green, 2021*). Overall, the use of ML with synthesis procedure datasets and chemical information has shown great promise for advancing our understanding of chemical processes and accelerating the discovery of new compounds with desired properties.

Text vectorization is a process by which texts are converted into numeric vectors, thus becoming suitable input for ML algorithms. The methods have evolved through several stages, starting from one-hot representation (*Al-Shehari & Alsowail, 2021*) and bag-of-words (BoW) (*Wu & Hoi, 2011*), moving to Word2Vec (*Mikolov et al., 2013*), GloVe (*Pennington, Socher & Manning, 2014*), FastText (*Bojanowski et al., 2017*), and SentencePiece (*Kudo & Richardson, 2018*). One-hot representation employs binary coding for generating word vectors, where each dimension indicates the presence or absence of a corresponding word from the pre-defined dictionary. BoW improves upon one-hot representation by incorporating information about word frequencies. However, both vectorization techniques produce discrete embeddings suffering from high dimensionality, data sparsity problems, and an inability to capture text semantics. Word2Vec, Glove, FastText, SentencePiece approaches vectorize words in the distributional representation manner: *i.e.*, produced word vectors contain real numbers and are of the predetermined fixed-length N. These vectorization approaches consider a small context window around the target word and thus learn to project words into N-dimensional space according to their similarity. FastText and SentencePiece learn to vectorize subwords (by providing a more flexible and granular representation of words) that are later used to compose words and create their vectors. Due to this additional mechanism, methods can vectorize morphologically rich texts and even out-of-vocabulary (OOV) words; moreover, their vectors are very similar to the correct equivalents. Although distributional vectorization techniques solve high dimensionality and data sparsity problems, they still cannot capture all the semantics as words written the same but have the same vectors that vectorize different meanings. On the contrary, word embeddings such as ELMo (*Peters et al., 2018*), BERT (*Wang et al., 2019*), and GPT (*Radford et al., 2018*) provide more accurate representations because words with multiple meanings are vectorized differently considering their larger context (within one or several sentences). These advanced text vectorization methods have significantly improved natural language processing tasks across various applications and domains. Despite many options, a common technique

used for chemical text remains SentencePiece: it uses a byte-pair-encoding algorithm as an effective mechanism to handle OOV words, such as chemical nomenclature.

Modern deep learning methods have significantly advanced in text classification and text generation fields, with various offered modifications and state-of-the-art architectures demonstrating high performance. Within this space, recurrent neural networks (RNNs) and long short-term memory (LSTM) (*Nowak, Taspinar & Scherer, 2017*) or gated recurrent units (GRU) are taking a significant position as methods adjusted to process sequences and, therefore, suitable for dealing with texts. Moreover, LSTMs and GRUs are improved versions of RNNs (no longer suffering from the vanishing gradient problem) with larger memory and are better adjusted to process longer sequences (*Cho et al., 2014*). The Transformer architecture has gained prominence due to its self-attention mechanism, enabling efficient modeling of long-range dependencies. Pre-trained language models (like BERT (*Devlin et al., 2018*), Roberta (*Liu et al., 2019*), XLNet (*Yang et al., 2019*)) or text-to-text transformers (like T5 (*Raffel et al., 2019*)) have revolutionized the field of natural language processing. These models are already pre-trained on large text corpora to understand language and, therefore, only need fine-tuning for specific tasks, usually demonstrating state-of-the-art performance across a wide range of benchmarks. BERT, RoBERTa, and XLNet are all bidirectional transformer models; BERT uses static masking with the following sentence prediction, whereas RoBERT only dynamic masking (where masked tokens change during training epochs); XLNet utilizes a permutation-based training strategy (where all tokens are predicted, but in random order). T5 extends the concept by formulating all NLP tasks (even classification) as text-to-text problems and employs a denoising autoencoder for pre-training. When fine-tuned on specific tasks, these models achieve state-of-the-art performance across various NLP benchmarks.

This article's primary contributions examine two critical tasks for extracting actions from chemistry patents. The first task (the text classification problem) isolates synthesis procedures within patent documents. We have developed a methodology that learns the model from the newly labeled dataset explicitly created for this purpose. The second task (the text generation problem) converts unstructured synthesis procedures into structured formats to extract compounds, actions, and action parameters. For this purpose, a deep learning-based methodology has been developed, surpassing previous work on text conversion in the number of parameters that can be extracted. In addition, all datasets and models are open-source, allowing for improvements and research applications. The models can be executed locally in a GPU environment, facilitating processes vital in the chemistry and cheminformatics fields due to substantial data volumes. We also present a novel dataset of synthesis procedures derived from USPTO and EPO patent data. This dataset employs the code and models discussed in the article and represents the largest open-source synthesis procedures dataset combining both data sources, containing 3,058,295 unique synthesis procedures in original and structured formats. Our research findings pave the way for chemists, AI researchers, and data scientists to tackle more complex tasks.

## The formal definition of the task

### Task 1–classification of paragraphs

Task 1 is a binary classification problem that aims to automatically classify paragraphs as either relevant or irrelevant to a given topic. Let $P$ be a set of paragraphs, where each paragraph $p \in P$ consists of a sequence of tokens (words, numbers). Let $T$ be a single target topic, representing the information we want to extract from the paragraphs. Let $Y = \{0, 1\}$ be a binary space of class labels, where $0$ represents irrelevant, and $1$ represents relevant paragraphs with respect to the topic $T$. Let $\eta$ be a mapping function $\eta(p) \to Y$ which, for each paragraph, can predict whether it is relevant or not to the topic $T$.

Let $\Gamma$ be an ML algorithm that could learn an approximation (denoted as $\eta'$) of function $\eta$ from the training dataset $D_P \subset P$. The goal of $\Gamma$ is to learn which model is able to predict, as accurately as possible, the class labels from their inputs automatically on the testing dataset $D_T$, $D_T = P - D_P$. The $D_P$ and $D_T$ datasets are not overlapping ($D_P \cap D_T = \phi$); both have enough diversity and are correctly distributed in the space. If both conditions are met, the evaluation results will be considered reliable. The objective of the task is to find the best possible mapping function $\eta$ that can accurately classify paragraphs as relevant or irrelevant to the topic $T$.

### Task 2–conversion of paragraphs

Given a source language sentence $s = (s_1, s_2,\ldots, s_n)$ in language $L_1$ (natural English language), the task is to generate a target language sentence $t = (t_1, t_2,\ldots, t_m)$ in language $L_2$ (a formal language with specific notation) that conveys the same meaning as the input sentence. Rather than mapping specific sentences from the input to the output, the process involves analyzing the input sentence word by word and generating the output sentence in a similar manner.

Let $S$ be the space of all possible source sentences in language $L_1$ and $T$ be the space of all possible target sentences in language $L_2$. Let $\Gamma$ be an ML algorithm that could learn a function $\eta(S) \to T$, which maps a source sentence to its corresponding target sentence.

The goal of $\Gamma$ is to learn an approximation (denoted as $\eta'$) of the function $\eta$ from a training dataset $D_S \subset S$, where each source sentence $s$ in $D_S$ has a corresponding target sentence $t$ in a target language $L_2$. The learned function $\eta'$ is evaluated on a separate testing dataset $D_T \subset S$, which consists of source sentences that have not been seen during the training phase. Finally, the model's performance is evaluated based on the accuracy and fluency of the generated target sentences.

## The data

### Task 1 dataset

In this study, we utilized a subset of the organic chemistry patent dataset from USPTO and EPO patents (as detailed in the Materials and Methods section) to create a dataset prepared explicitly for ML algorithms employed in Task 1. The instances were manually labeled for the classification task, resulting in a dataset comprising 20,199 paragraphs with binary labels. Each paragraph is categorized as either an organic synthesis procedure (labeled *1*) or a non-synthesis paragraph from patent documents (labeled *0*). A paragraph is considered

accurate for a procedure if it explicitly outlines the chemical compounds, actions, and action parameters involved in chemical synthesis. Generally, patents encompass 5 to 80 or more paragraphs, with a significant portion dedicated to compound usage processes, new drug formulations, chemical and physical properties of compounds, laboratory setups, devices, and in-depth chemical reaction mechanisms. Approximately 5–7% of all paragraphs within a patent specifically describe chemical reaction procedures, including the chemical compounds, steps, and actions involved.

Synthesis procedure paragraphs typically follow a consistent structure, starting with the description of reactants and reaction conditions, followed by chemical manipulations such as quenching, crystallization, filtration, purification, and ultimately naming the resulting compound along with its yield. While complexity may vary, the general structure is maintained. Several criteria have been established to minimize the occurrence of irregular procedures. For instance, a procedure description should contain at least 10 words, as no shorter examples were identified. Procedure descriptions should not end abruptly, as there are cases where, for unknown reasons, a procedure is only a fragment of the complete one. Additionally, procedure instructions must not reference methods or procedures, *e.g.*, *Following general procedure A* or *as described in*. Finally, compounds should not be referenced in the procedure description, such as *Compound 1* or *INTERMEDIATE 2*. Instead, each compound must adhere to the International Union of Pure and Applied Chemistry (IUPAC) chemical nomenclature. This requirement is particularly relevant to product names, as approximately 10–20% of procedures do not explicitly state the product name, referring to it with terms like *title compound, resultant compound, white solid, yellow crystals, or black powder*. The product may be referenced if the proper compound name is also provided. Procedures that did not meet these criteria were considered non-organic synthesis procedures and were labeled accordingly.

The dataset comprises 20,199 instances, divided into training (80%, 16,159), validation (10%, 2,020), and testing (10%, 2,020) subsets. Each instance contains an input-output pair, with the input being a sentence or paragraph and the output being a binary label represented as one or zero. The dataset features a wide range of sentence lengths, with the shortest being 11 words and the longest being 1,053 words. Despite this range, the average sentence length is 86.93 words overall, with 87.08 words for the training subset, 85.13 for the testing subset, and 87.5 for the validation subset. The subsets were randomly selected, and their similar sentence lengths ensured reliable results during testing. The output labels are unevenly distributed, with 15,114 zeros and 5,085 ones, corresponding to 33.6% of the total instances. The training, validation, and testing subsets maintain a similar ratio between the two classes.

The datasets will be used for supervised learning algorithms, and the models must be compared to random and majority baselines. To be considered appropriate for our problem-solving task, the method's accuracy must surpass both the random baseline, which denotes the minimum accuracy required to differentiate the method from a random labeler, and the majority baseline, which indicates the accuracy that would be achieved if all instances were assigned to the most frequent class.

$$Random\ baseline = \sum_{i=1}^{n} (P(y_i))^2 \tag{1}$$

$n$—number of classes, $(P(y_i))$–the probability of $y_i$ class.

$$Majority\ baseline = \max(P(y_i)) \tag{2}$$

The calculated random and majority baselines for both datasets equal 0.553 and 0.664.

### TASK 2 dataset

The primary objective of Task 2 is to develop a method for converting unstructured synthesis procedure text into a structured format. For this purpose, we have selected supervised ML algorithms requiring a dataset of input-output example pairs. The input data comprises sentences from synthesis procedures, while the output data are their structured counterparts. A structured sentence simplifies the original by representing the action with a single word, followed by specific action parameters. The possible actions include *Add, CollectLayer, Concentrate, Degas, DrySolid, DrySolution, Extract, Filter, FollowOtherProcedure, MakeSolution, Microwave, OtherLanguage, Partition, PH, PhaseSeparation, InvalidAction, Purify, Quench, Recrystallize, NoAction, Reflux, SetTemperature, Sonicate, Stir, Triturate, Wait, Wash,* and *Yield.* Each action name streamlines a sentence, making it more human-readable and understandable. We use the schema for the names of the actions proposed by IBM because it is currently the most comprehensive for this task (*Vaucher et al., 2020*). A detailed explanation of each step can be found in Appendix S1. Training ML models for tasks involving text translation typically requires large datasets. Consequently, two datasets are employed for this task: the pre-training and annotated datasets, which are described in greater detail in a subsequent section. Table 1 provides examples from the dataset.

The pre-training dataset was created using *Lowe's (2017)* and NextMove's open-source collection of chemical reactions extracted from US patents issued between 1976 and 2016, which served as the primary source of synthesis procedure paragraphs. Initially, paragraphs were split into sentences using the SPACY (https://spacy.io, accessed 18 March 2022) natural language processing library. Subsequently, the IBM RXN API (https://rxn.res.ibm.com, accessed 22 April 2022) was employed to convert the sentences into a structured format. The obtained data is licensed under CC-BY (https://creativecommons.org/licenses/by/4.0/). In total, 4,994,532 examples were collected over 7 months. Some examples were discarded because the sentences were only a few words long, contained just numbers, or were incomplete sentences. In addition, because the dataset was generated by a neural network model without human annotation, it may contain noise. To assess data quality, 300 randomly selected examples were sampled, corrected, and evaluated using the BLEU metric (4-gram with uniform weights), yielding a score of 0.913. Although the score was relatively high, the pre-training dataset is a silver standard. The dataset's size allows for the initial training stage of ML models. It is composed of 3,074,038 instances, with input-output pairs where the input corresponds to a sentence. The minimum original sentence

**Table 1 An excerpt from the dataset presents five examples of input and output pairs.**

| Original sentence | Structured format |
|---|---|
| To a solution of 3-cyano-4-((1-methylethyl)oxy)benzoic acid (200 mg) in tetrahydrofuran (THF) (10 mL) was added EDC (374 mg) and HOBt (299 mg). | MAKESOLUTION with 3-cyano-4-((1-methylethyl)oxy)benzoic acid (200 mg) and tetrahydrofuran (THF) (10 mL); ADD SLN; ADD EDC (374 mg); ADD HOBt (299 mg). |
| The resulting orange solution was partitioned between dichloromethane (10 mL) and water (10 mL) | PARTITION with dichloromethane (10 mL) and water (10 mL). |
| Ethanol and water are added to produce a solid which is collected by filtration. | ADD Ethanol; ADD water; FILTER keep precipitate. |
| The solution was quenched with saturated aqueous NH4Cl and extracted with CH2Cl2. | QUENCH with saturated aqueous NH4Cl; EXTRACT with CH2Cl2. |
| The resulting mixture was filtered and the filtrate was concentrated and the residue was purified by flash column chromatography (ethylacetate/petroleum ether gradient) to afford 1-benzyl-2-(chloromethyl)pyrrolidine. | FILTER keep filtrate; CONCENTRATE; PURIFY: ethyl acetate:petroleum ether; YIELD 1-benzyl-2-(chloromethyl)pyrrolidine. |

length is one word, while the maximum is 850. On average, sentences are 22.6 words long overall, with 22.7 (training subset), 22.6 (testing subset), and 22.4 (validation subset). The subsets were randomly selected, maintaining similar sentence lengths for reliable testing results. The output consists of a structured version of a synthesis procedure sentence. The dataset was randomly divided into training (80%, 2,459,230), validation (10%, 307,404), and testing (10%, 307,404) subsets. The frequency of the action was observed to be very similar among the subsets. Although synthesis action parameters can vary considerably, the frequency of actions is observable, as shown in Table 2. Note that a single sentence may contain multiple actions, so the total number of actions may exceed the number of instances.

The pre-training dataset was further enhanced by incorporating parameters for the *Purify* action. In addition, a custom script was developed to extract information about solvents and their ratios. Initially, the structured representation lacked information on the *Purify* action, but the pre-training dataset was later augmented with the action's parameters. The additional parameters describe the ratio and the solvents used for the purification process.

The annotated *gold standard* dataset was created using a portion of synthesis procedures not included in the pre-training dataset and was annotated by an experienced chemist. This dataset was also enriched with the *Purify* action parameters. Naturally, the annotated dataset is significantly smaller, with 1,008 instances. It has been divided into training (80%), validation (10%), and testing (10%) subsets. The average length of input sentences in the pre-training and annotated datasets was similar (pre-training–22.6, annotated–22.06). The two datasets' action frequency was also comparable, indicating their similarity. The annotated dataset was employed in the second stage of model training, while the testing dataset was used to assess the final performance metrics.

In this study, we employ various data augmentation techniques to expand the datasets. First, the sentences are converted from the original to a structured version. Data augmentation is used in this chemistry-related task to increase the diversity and size of the

**Table 2 The total number of each action in the pre-training dataset.**

| Action name | Total count | Action name | Total count |
| --- | --- | --- | --- |
| *Add* | 2,249,260 | *Filter* | 352,100 |
| *Stir* | 671,005 | *DrySolution* | 336,816 |
| *Concentrate* | 656,695 | *Purify* | 323,716 |
| *Yield* | 618,280 | *SetTemperature* | 275,622 |
| *Wash* | 506,832 | *Extract* | 245,546 |
| *MakeSolution* | 476,329 | *NoAction* | 207,310 |
| *CollectLayer* | 364,382 | *FollowOtherProcedure* | 168,541 |
| *Reflux* | 102,683 | *InvalidAction* | 44,904 |
| *PH* | 95,297 | *Partition* | 34,534 |
| *PhaseSeparation* | 67,165 | *Triturate* | 29,702 |
| *Wait* | 65,751 | *Degas* | 24,100 |
| *DrySolid* | 64,692 | *Microwave* | 11,988 |
| *Recrystallize* | 62,959 | *OtherLanguage* | 1,936 |
| *Quench* | 49,396 | *Sonicate* | 1,447 |

training data, which can help improve the model's generalization and robustness to variations in input. The augmentation is achieved in four different aspects: (1) substituting temperature parameters in both the original and structured versions with randomly selected numbers and temperature notations from the original dataset (*e.g., C, Celsius*); (2) replacing duration parameters with randomly chosen numbers and corresponding units (*e.g., hours, s*); (3) modifying solvents used in actions such as *Wash* or *Extract* with randomly selected solvents or combinations of solvents and (4) exchanging compound names with alternative compounds. To maximize efficiency, we implement these augmentation methods in conjunction, utilizing extensive lists of temperature and duration notations, compounds, and solvents available in the project repository on GitHub (https://github.com/Mantas-it/ActionExtraction). We apply these data augmentation techniques to both the pre-training and annotated datasets, limiting each instance's augmentation to four times to maintain a manageable training dataset size and reasonable use of processing resources. The pre-training dataset has 15,503,534 instances, while the fine-tuning dataset has 19,642 instances.

# MATERIALS AND METHODS

## Vectorization

Vectorization is a crucial technique that involves transforming raw data into a numerical form that can be fed into ML algorithms. Vectorization, in our case, is necessary as we use neural networks that rely on mathematical operations to process the data and make predictions. Our research objective is texts (which we have in both tasks' input and output for Task 2). The section below describes each task's applied vectorization process in more detail.

## Vectorization for Task 1

The difficulty of vectorizing text about organic chemistry is because it contains many rare or unseen terms, such as newly discovered compounds or experimental results, molecular formulas, and chemical reactions that are not included in existing vocabularies (*Perera, Dehmer & Emmert-Streib, 2020*). The problem is generally called an open vocabulary problem and refers to the challenge of dealing with an infinite or unknown set of possible inputs. Open vocabulary problems can pose significant challenges for ML models, requiring them to handle previously unseen inputs that may not conform to known patterns or rules (*Eichstaedt et al., 2021*; *Mielke et al., 2021*). It also automatically excludes standard closed vocabulary methods, such as Bag-of-Words (BoW) (*Qader, Ameen & Ahmed, 2019*) or one-hot encoding (*Zhang & LeCun, 2017*). One possible approach is to use SentencePiece tokenizer, a method that breaks down the text into smaller subunits, such as individual characters or word pieces, and then represents these subunits as numerical vectors. It is also consistent with the fact that chemical nomenclature is composed of word fragments that describe specific functional groups and the sub-structure of the molecule. For this reason, the SentencePiece tokenizer has been selected as one of the most appropriate for the project (*Mugisha & Paik, 2022*; *Abdel-Aty & Gould, 2022*). Another possibility would be to use character-level embeddings when text is represented as a sequence of vectors, where each vector corresponds to a character in the text (*Gajendran & Sugumaran, 2020*). However, vectorizing long paragraphs can be computationally expensive and may not be the most efficient way to represent text data, so it is not considered in our research article.

Preparing a SentencePiece tokenizer starts with collecting the entire *corpus* of text data, preprocessing by removing any non-textual elements, such as HTML tags or special characters, and converting the text to lowercase if necessary. The sentencepiece library was used in our research article. Next, the text is segmented into variable-length subword units using unigram language modeling (*Park et al., 2021*). The subword units are then sorted based on their frequency of occurrence in the *corpus*, and a predefined number of units are selected to form the final vocabulary. Next, the training process starts, and the subword segmentation and vocabulary generation steps are repeated iteratively on the *corpus* until convergence to maximize the likelihood of the observed text data given the subword vocabulary. The SentencePiece algorithm has several essential parameters that can be customized for optimal performance. This research tested six vocabulary sizes, *i.e.*, 2,500, 5,000, 10,000, 16,000, 32,000, and 64,000. The larger vocabulary can potentially improve the accuracy (because even rare subwords can be recognized and vectorized), but it also requires more computational resources. The algorithm can also split digits so that individual digits and corresponding numerical integers encode each number. However, texts with formulas and complex nomenclature often have several numbers in a single sentence. Therefore, it is advisable not to vectorize with split digits to preserve the size of the final vector. The process generates a vocabulary of set size which then was used to vectorize the input data. The output is a binary label and does not require any additional processing.

## Vectorization for Task 2

For the second task, we used the same vectorization technique, *i.e.*, the SentencePiece tokenizer, to vectorize both input and output data. This method was chosen because, similarly to the first task, the input data is in text form and contains several unique and domain-specific terms related to organic chemistry procedures, compounds, and solvent names. The output data is also text-based but structured, representing actions and parameters of actions. Therefore, a suitable vectorization method was necessary to represent the text data in a way ML methods could process. The vocabulary of input and output is almost identical; therefore, the SentencePiece model has been trained from the entire dataset. Due to it, shared embedding layers with Transformers may be used for ML methods. The SentencePiece vectorization method effectively and efficiently represents the input and output data for translating organic chemistry procedure sentences into structured sentences.

## Deep learning methods

Deep learning (DL) is a type of ML that has gained significant attention in recent years due to its ability to approximate the relationships between input and output data. DL models are composed of multiple layers of interconnected units, which can learn hierarchical representations of the input data. The DL has applications in various research fields, including computer vision, natural language processing, and drug discovery. However, the effectiveness of DL methods can vary significantly depending on various factors, such as the nature of the solving task, dataset diversity, and other important characteristics. In the context of our research, we aim to classify and convert text sequences, so it is essential to carefully consider the types of methods used to train DL models. In the following section, we present the most suitable ones for our solving tasks.

## Deep learning methods for Task 1

For classifying patent paragraphs into chemical procedures or other descriptions, we explored the following types of ANNs:

- **GRU** (gated recurrent unit) is a type of recurrent neural network introduced to address some of the limitations of standard RNNs in processing sequential data. One of the main limitations of standard RNNs is the vanishing gradient problem (*Rehmer & Kroll, 2020*), which can make it difficult for the network to learn long-term dependencies in the data. GRUs use gated units to control the flow of information through the network. However, GRUs have few parameters, making them more computationally efficient and easier to train. GRUs also have a relatively simple architecture, making them a good choice for modeling short-term dependencies in the data. In particular, the reset and update gates in GRUs allow the network to selectively forget or update information from the previous time step, which can help model sequential data (*Gruber & Jockisch, 2020*). Therefore, GRUs have been chosen as the most basic type of architecture for this task.

- **LSTM** (long short-term memory) is a recurrent neural network for handling sequential data with long-term dependencies. Similarly, to RNNs, LSTMs have a short-term memory connected to weights and biases, but unlike standard RNNs, LSTMs have

cell states where memories flow through time steps and do not affect the gradient. The memory cell in LSTMs has three gates that allow it to selectively add, remove, or retain information at each time step. These gates are known as the input gate, forget gate, and output gate (*Staudemeyer & Morris, 2019*). The input gate determines how much new information is added to the memory cell at each time step, the forget gate decides what information is removed from the memory cell, and the output gate controls how much of the memory cell is used to generate the output at each time step.

• **BiLSTM** (bidirectional LSTM) is a variation of LSTM that processes input data in both forward (from past to future) and backward (from future to past) directions (*Alawneh et al., 2020*). In addition, BiLSTMs have mechanisms allowing to merge models of both input directions, making this feature particularly useful for tasks where context and dependencies from both directions are important.

• **Transformer**. The traditional transformer neural network architecture (*Lakew, Cettolo & Federico, 2018*; *Shao et al., 2019*) consists of the encoder and decoder blocks and relies on attention mechanisms. Self-attention (used in encoder and decoder blocks) allows the model to better "understand" how tokens in the sequence depend on each other. This is done by computing a set of attention scores between each token in the sequence and every other token, which are then used to compute a weighted sum of the input embeddings. The encoder-decoder attention is responsible for focusing on the appropriate parts in the input sequence when making the predictions (*Liu et al., 2018*). In addition to attention, the transformer also uses residual connections and layer normalization, which help mitigate the vanishing gradient problem and improve the stability of the training process. For this task, we trained the transformer model from scratch, which consists of an encoder connected to the feed-forward layer. Afterward, the feed-forward layer is connected to the output layer, producing a single numerical value for binary classification. Finally, each model was explored to determine the best fit for the classification task.

## Deep learning methods for Task 2

For converting sentences of chemical procedures to structured and simplified formats, we explored the following types of ANNs.

• **Seq2seq LSTM** is a type of neural network architecture designed to learn mappings from an input sequence to an output sequence of variable length. This is achieved using two separate LSTMs: an encoder network that processes the input sequence and generates a fixed-length context vector and a decoder network that uses the context vector to generate the output sequence.

• **Seq2seq BiLSTM**. Similarly to seq2seq LSTM, this approach used BiLSTMs instead of LSTMs by processing the input in the encoder in both directions (*Ro et al., 2022*; *Egonmwan & Chali, 2019*). We anticipate that the bidirectional aspect of the model should better capture the context and dependencies in the input sequence, which can lead to improved performance. The decoder layer contained LSTM.

• **Transformer**. We have used the traditional transformer architecture adjusted for our seq2seq problem (*Chi et al., 2021*; *Jawahar et al., 2021*), allowing input and output sequences of different lengths, such as used in, *e.g.*, machine translation (*Garg et al., 2021*;

*Senadeera & Ive, 2022*). In this architecture, the encoder takes the input sequence and generates a series of context vectors, one for each time step in the input sequence. The decoder then takes these context vectors as input and generates the output sequence one token at a time. At each time step, the decoder uses an attention mechanism to focus on the relevant parts of the input sequence, allowing it to generate an output token informed by the entire input sequence. One of the key advantages of this architecture is its ability to handle long-range dependencies in data, allowing it to capture complex patterns in sequences of arbitrary length. We have utilized the model described in an article by *Vaswani et al. (2017)* for this task. An open-source Python library OpenNMT-tf was utilized to implement the tokenization algorithms and training.

- **T5** (text-to-text transfer transformer) architecture is one of the most advanced in the field of natural language processing (NLP), offering an adaptable approach to various tasks (*Mars, 2022*). T5 employs an encoder-decoder architecture like other seq2seq transformer models. The encoder processes input text and generates a series of context vectors, while the decoder attends to these context vectors to produce the output text (*Hui et al., 2022*). Incorporating self-attention mechanisms enables the model to effectively capture long-range dependencies and relationships between words in input and output sequences. An essential aspect of the T5 architecture is implementing a masked language modeling technique during the pre-training phase (*Young & You, 2023*; *Wettig et al., 2022*). This method involves masking specific input text segments, and the model predicts the omitted words. As a result, the model acquires an understanding of language structure and semantics. A pre-trained model is often used to fine-tune a particular task to bypass the computationally intense task of training the model from scratch. Its capacity to capture long-range dependencies and understand complex language patterns makes it an optimal choice for numerous NLP applications (*Bird, Ekárt & Faria, 2021*; *Najafi & Tavan, 2022*), such as text translation for our task. We have utilized the *HuggingFace* library for this task: base model-(https://huggingface.co/t5-base, accessed 12 December 2022), small model-(https://huggingface.co/t5-small, accessed 12 December 2022).

## Hyper-parameters and optimization

After correctly defining the task (see "The formal definition of the task" section) and choosing promising types of methods (see the "Materials & Methods" section), the choices of their hyper-parameter values are no less important. Moreover, the optimal values of hyper-parameters play a crucial role in the model training process (*Yu & Zhu, 2020*) as they directly affect the model's evaluation results. In this research, we have investigated the following hyper-parameters and their values:

- Activation functions (investigated values: ReLU, GELU, SELU, ReLU, ELU, and tanH) (*Hendrycks & Gimpel, 2016*; *Rasamoelina, Adjailia & Sincak, 2020*). Activation functions determine how the model will process and transform input data.
- Optimizer (Adam, Nadam, SGD, Adamax, and RMSprop) (*Bischl et al., 2023*). The optimizer is responsible for updating the model weights during training, and different optimizers can affect the speed and quality of the learning process.

• Batch size (16, 32, 64, 128, 256, 512). The batch size determines how many samples are processed in each training process iteration. Therefore, different batch sizes can affect the speed and quality of the learning process (*Smith et al., 2017*).

• Attention heads (2, 4, 6, 8) (for Transformer only) Varying the number of attention heads impacts the model's capacity to process information efficiently, with a trade-off between performance and computational resources.

• Neural network layer size (16, 32, 64, 128, 256, 512) Layer size influences the model's complexity and expressive power, with larger sizes potentially offering better performance at the cost of increased computational resources and overfitting risk.

Hyperparameter optimization in deep learning models involves systematically searching and selecting optimal hyperparameters. This process used a grid search technique to navigate the hyperparameter space and evaluate various configurations. The goal is to identify the hyperparameter combination that minimizes the model's generalization error and maximizes its performance on unseen data. We have utilized a technique of early stopping to stop the training process if the validation accuracy did not increase in the seven latest epochs. All models used a loss function of binary cross-entropy in the output layer. We used the *wanDB* platform (www.wandb.ai, accessed on 6 September 2022), which provides visualization of multiple runs and is convenient for navigation and analysis. The following hyper-parameters were optimized:

## Optimized parameters for Task 1

Neural network layer size: 16, 32, 64, 128, 256, 512.
Embedding dimensions: 4, 6, 8, 16, 32.
Activation functions: GELU, SELU, ReLU, ELU, and tanH.
Optimizers: Adam, Nadam, SGD, Adamax, and RMSprop.
Dropout: an interval of 0.1 to 0.8.
Sample importance factor: interval of 0.5 – 3.0.
Batch sizes: 32, 64, 128, 256.
Only for transformer architecture. The number of heads: 2, 4, 6, 8 and the number of layers 1, 2, 3, 4.

## Optimized parameters for Task 2

Neural network layer size: 16, 32, 64, 128, and 256.
Embedding dimensions: 4, 6, 8, 16 and 32.
Activation functions: GELU, SELU, ReLU, ELU, and tanH.
Optimizers: Adam, Nadam, SGD, Adamax, and RMSprop.
Batch sizes: 32, 64, 128, 256.
Transformer architectures were not additionally optimized due to a significantly higher computational cost, and only different vocabulary sizes were tested. A pre-trained T5 model was used, and therefore hyper-parameters were not changed.

**Table 3 Optimal hyper-parameters for GRU, LSTM, and BiLSTM neural networks architectures for Task 1.**

|  | Vocabulary size | Embedding dimension | Dropout | LSTM/GRU layer size | FF layer size | Importance | Optimizer |
|---|---|---|---|---|---|---|---|
| GRU | 2,500 | 8 | 0.199 | 32 | 8 | 1.282 | RMSprop |
|  | 5,000 | 8 | 0.367 | 4 | 32 | 1.016 | RMSprop |
|  | 10,000 | 8 | 0.681 | 64 | 256 | 1.228 | RMSprop |
|  | 16,000 | 8 | 0.677 | 8 | 8 | 1.046 | Nadam |
|  | 32,000 | 8 | 0.649 | 4 | 256 | 1.042 | Nadam |
|  | 64,000 | 8 | 0.593 | 4 | 64 | 1.195 | Adamax |
| LSTM | 2,500 | 8 | 0.340 | 8 | 256 | 2.423 | Nadam |
|  | 5,000 | 8 | 0.650 | 16 | 16 | 1.268 | Nadam |
|  | 10,000 | 8 | 0.594 | 16 | 64 | 1.598 | Adam |
|  | 16,000 | 8 | 0.189 | 64 | 8 | 1.055 | Adamax |
|  | 32,000 | 8 | 0.643 | 64 | 32 | 2.187 | Adam |
|  | 64,000 | 8 | 0.207 | 64 | 8 | 2.173 | Adam |
| BiLSTM | 2,500 | 6 | 0.646 | 8 | 256 | 1.519 | Adamax |
|  | 5,000 | 8 | 0.678 | 16 | 8 | 1.622 | RMSprop |
|  | 10,000 | 8 | 0.405 | 8 | 8 | 1.860 | Adam |
|  | 16,000 | 8 | 0.639 | 16 | 8 | 2.493 | RMSprop |
|  | 32,000 | 8 | 0.279 | 16 | 16 | 1.039 | Adamax |
|  | 64,000 | 8 | 0.430 | 16 | 16 | 1.870 | RMSprop |

## Optimal method architectures and hyper-parameters

The models were trained to employ the most suitable parameters and architectures identified through the optimization process for evaluation. This section describes the optimal parameters and respective architectures for both tasks. The GitHub repository (https://github.com/Mantas-it/ActionExtraction) provides access to the corresponding models and their optimal parameter combinations.

## Summary of optimal hyper-parameters for Task 1

The GRU, LSTM, and BiLSTM neural network architectures are composed of the embedding layer, a dropout layer, a corresponding GRU/LSTM/BiLSTM layer, and a feed-forward (FF) layer connected to the output in sequence. Table 3 illustrates the optimal hyper-parameters for all combinations with different vocabulary sized. The optimal activation function of the FF layer was found to be ReLU in all cases. Similarly, a batch size of 64 led to the best performance and is not displayed in the table. The embedding dimension represents the number of dimensions the embedding layer uses. The importance value signifies how much more important are the samples that belong to the class of real procedures because the training dataset is not balanced.

The transformer neural networks for Task 1 consist of the embedding layer and one or more blocks of multi-head attention layer combined with the normalization layer. Table 4 presents relevant hyper-parameters for various vocabulary sizes. The dropout was set to

**Table 4 Optimal hyper-parameters for transformer neural networks architectures for Task 1.**

|  | Vocabulary size | Embedding dimension | Batch size | Number of heads | Attention layers | Importance | Optimizer |
|---|---|---|---|---|---|---|---|
| **Transformer** | 2,500 | 16 | 32 | 2 | 1 | 1.245 | RMSprop |
|  | 5,000 | 8 | 128 | 4 | 2 | 1.928 | RMSprop |
|  | 10,000 | 8 | 64 | 8 | 4 | 1.594 | Adam |
|  | 16,000 | 8 | 32 | 2 | 2 | 1.117 | Adamax |
|  | 32,000 | 8 | 64 | 8 | 4 | 2.226 | RMSprop |
|  | 64,000 | 8 | 64 | 4 | 4 | 1.764 | RMSprop |

**Table 5 Optimal hyper-parameters for seq2seq LSTM and BiLSTM architectures for Task 2.**

|  | Vocabulary size | Embedding dimension | LSTM/BiLSTM encoder size | LSTM/BiLSTM decoder size | Optimizer |
|---|---|---|---|---|---|
| **seq2seq LSTM** | 8,000 | 32 | 256 | 256 | Adam |
|  | 16,000 | 64 | 128 | 128 | Adamax |
|  | 32,000 | 64 | 128 | 128 | RMSprop |
| **seq2seq BiLSTM** | 8,000 | 64 | 256 | 128 | Adamax |
|  | 16,000 | 64 | 256 | 128 | Adam |
|  | 32,000 | 64 | 128 | 128 | Adam |

zero because higher values led to an unstable training process. The attention layers number describes how many multi-head attention layers were used.

## Summary of optimal hyper-parameters for Task 2

The seq2seq LSTM and BiLSTM architectures are constructed by connection encoder and decoder, which themselves are made up of an input layer, an embedding layer, and a corresponding LSTM or BiLSTM layer. Table 5 describes the hyper parameters that were found to be optimal for different vocabulary sizes. A batch size of 64 was used for the training process to conserve memory use.

The transformer model for the translation task was implemented *via* the *OpenNMT-tf* library (Materials & Methods). Although the models were only trained with a vocabulary of different sizes, the model architecture was not optimized. The model consists of an encoder and a decoder, each with six stacks. Each stack consists of a multi-head attention layer (eight heads and a size of 512), a normalization layer, and a feed-forward layer (of size 2,048). Other values were left as default, and only the learning rate (0.005), batch size (4,096), and effective batch size (32,768) were modified. The T5 small and base models (Materials & Methods) were only fine-tuned and were not subjected to hyperparameter optimization.

## RESULTS

The following experiments were performed for two tasks (see "The formal definition of the task" section), and the results of the testing dataset are presented in the tables below. In

addition, each model has been trained and evaluated three times with optimal hyper-parameter values (Materials & Methods).

## Results of Task 1

As discussed in the Methods & Materials section, the binary classification task was solved, and its performance was evaluated using four key metrics: Accuracy (Eq. (3)), Precision (Eq. (4)), Recall (Eq. (5)), and F1-score (Eq.(6)). The true positive (*TP*) represents instances where $y_i$ was accurately predicted as $y_i$; true negative (*TN*) corresponds to cases where $y_j$ was correctly identified as $y_j$; false positive (*FP*) refers to instances where $y_j$ was erroneously predicted as $y_i$; and false negative (*FN*) denotes cases where $y_i$ was inaccurately predicted as $y_j$.

$$Accuracy = \frac{TP + TN}{TP + FP + TN + FN} \tag{3}$$

$$Precision = \frac{TP}{TP + FP} \tag{4}$$

$$Recall = \frac{TP}{TP + FN} \tag{5}$$

$$F1 - score = \frac{2 \times precision \times recall}{precision + recall} \tag{6}$$

For our evaluation, we have used precision, recall, and f1-score values. The results are averaged from three runs and presented in Table 6.

## Results of Task 2

As elaborated in the Methods & Materials section, we assessed the performance of our text generation task using two key metrics: BLEU (Bilingual Evaluation Understudy) (*Papineni et al., 2001*) and ROUGE-L (Recall-Oriented Understudy for Gisting Evaluation-Longest Common Subsequence) (*Lin, 2004*). These metrics are commonly used to evaluate the quality of generated text compared to reference texts.

The BLEU score calculates the geometric mean of modified *n-gram* ($N = 4$) precision ($P_n$) with a brevity penalty (*BP*) to account for differences in length between the generated and reference texts:

$$BLEU = BP \times exp\left(\sum\left(\frac{\log(Pn)}{N}\right)\right) \tag{7}$$

where $N$ is the maximum order of *n-grams* considered, the brevity penalty (*BP*) is defined as:

$$BP = \begin{cases} 1 & if\ c > r \\ e^{(1-r/c)} & if\ c \leq r \end{cases} \tag{8}$$

where r is the total length of the reference texts, and c is the total length of the generated texts.

**Table 6 Evaluation results (averaged with calculated confidence intervals (95% confidence)) of Task 1 using the testing dataset.** The best results are presented in bold.

| | | Vocabulary size | | | | | |
|---|---|---|---|---|---|---|---|
| | | 2,500 | 5,000 | 10,000 | 16,000 | 32,000 | 64,000 |
| **GRU** | Accuracy | 0.920 ± 0.010 | 0.924 ± 0.013 | 0.920 ± 0.010 | 0.912 ± 0.006 | 0.912 ± 0.026 | 0.911 ± 0.020 |
| | Precision | 0.911 ± 0.007 | 0.910 ± 0.006 | 0.911 ± 0.012 | 0.893 ± 0.006 | 0.896 ± 0.002 | 0.882 ± 0.004 |
| | Recall | 0.889 ± 0.015 | 0.903 ± 0.022 | 0.899 ± 0.020 | 0.889 ± 0.026 | 0.890 ± 0.005 | 0.903 ± 0.008 |
| | F1-score | 0.900 ± 0.006 | 0.907 ± 0.013 | 0.905 ± 0.028 | 0.891 ± 0.004 | 0.893 ± 0.001 | 0.892 ± 0.018 |
| **LSTM** | Accuracy | 0.905 ± 0.001 | 0.909 ± 0.020 | 0.908 ± 0.001 | 0.900 ± 0.026 | 0.925 ± 0.006 | 0.903 ± 0.024 |
| | Precision | 0.873 ± 0.024 | 0.893 ± 0.023 | 0.895 ± 0.018 | 0.871 ± 0.018 | 0.907 ± 0.025 | 0.873 ± 0.010 |
| | Recall | 0.902 ± 0.022 | 0.880 ± 0.016 | 0.884 ± 0.010 | 0.885 ± 0.028 | 0.917 ± 0.027 | 0.894 ± 0.007 |
| | F1-score | 0.886 ± 0.006 | 0.886 ± 0.019 | 0.889 ± 0.025 | 0.878 ± 0.002 | 0.912 ± 0.007 | 0.883 ± 0.010 |
| **BiLSTM** | Accuracy | 0.761 ± 0.026 | 0.899 ± 0.001 | 0.919 ± 0.018 | **0.939 ± 0.005** | 0.928 ± 0.006 | 0.932 ± 0.012 |
| | Precision | 0.721 ± 0.028 | 0.891 ± 0.026 | 0.901 ± 0.024 | **0.934 ± 0.004** | 0.914 ± 0.030 | 0.929 ± 0.005 |
| | Recall | 0.789 ± 0.009 | 0.860 ± 0.018 | 0.907 ± 0.010 | **0.933 ± 0.005** | 0.912 ± 0.010 | 0.906 ± 0.003 |
| | F1-score | 0.732 ± 0.011 | 0.874 ± 0.020 | 0.904 ± 0.026 | **0.932 ± 0.010** | 0.913 ± 0.011 | 0.917 ± 0.021 |
| **Transformer** | Accuracy | 0.917 ± 0.004 | 0.917 ± 0.021 | 0.895 ± 0.011 | 0.917 ± 0.026 | 0.917 ± 0.024 | 0.903 ± 0.019 |
| | Precision | 0.903 ± 0.015 | 0.903 ± 0.004 | 0.876 ± 0.008 | 0.903 ± 0.009 | 0.903 ± 0.016 | 0.893 ± 0.016 |
| | Recall | 0.893 ± 0.012 | 0.893 ± 0.028 | 0.859 ± 0.003 | 0.893 ± 0.005 | 0.893 ± 0.009 | 0.888 ± 0.015 |
| | F1-score | 0.898 ± 0.030 | 0.898 ± 0.023 | 0.867 ± 0.026 | 0.898 ± 0.024 | 0.898 ± 0.022 | 0.894 ± 0.013 |

The ROUGE-L score measures the longest common subsequence (LCS) between the generated text and the reference text, taking into account recall and precision. The ROUGE-L score is calculated as follows:

$$ROUGE - L = \frac{\left(1 + \beta^2\right) \times LCS\ Precision \times LCS\ Recall}{\left(\beta^2 \times LCS\ Precision + LCS\ Recall\right)} \tag{9}$$

where $\beta = 1$ is a weighting factor that adjusts the importance of precision relative to recall. LCS Precision is the ratio of the length of the LCS to the length of the generated text, while LCS Recall is the ratio of the length of the LCS to the length of the reference text.

The results are presented in Table 7.

## Data processing methods for the EPO/USPTO dataset

This section delineates the methodology and steps employed in constructing a dataset comprising synthesis procedures, their structured counterparts, reactants, and products represented in IUPAC and SMILES notations, similar to the D. M. Lowe's dataset. Given the dataset's potential value for chemists and AI researchers, describing the process in detail is crucial. The procedure extends beyond conventional text preprocessing techniques and requires more steps and data preparation methods. The two main tasks analyzed in the article are included in the description, emphasizing that the outlined steps serve as one possible approach, which may be adapted or enhanced to align with the specific objectives

**Table 7 Evaluation results (averaged with calculated confidence intervals) of Task 2 using the testing dataset.** The best results are presented in bold.

| | | Vocab size | | |
| --- | --- | --- | --- | --- |
| | | **8,000** | **16,000** | **32,000** |
| **LSTM** | BLEU | 0.560 ± 0.009 | 0.574 ± 0.006 | 0.592 ± 0.006 |
| | ROUGE | 0.596 ± 0.011 | 0.602 ± 0.005 | 0.623 ± 0.007 |
| **BiLSTM** | BLEU | 0.612 ± 0.009 | 0.649 ± 0.003 | 0.671 ± 0.011 |
| | ROUGE | 0.635 ± 0.015 | 0.673 ± 0.008 | 0.695 ± 0.005 |
| **Transformer** | BLEU | 0.912 ± 0.011 | **0.946 ± 0.006** | 0.927 ± 0.011 |
| | ROUGE | 0.932 ± 0.014 | **0.950 ± 0.010** | 0.949 ± 0.004 |
| **T5 small** | BLEU | – | – | 0.906 ± 0.005 |
| | ROUGE | – | – | 0.928 ± 0.009 |
| **T5 base** | BLEU | – | – | 0.902 ± 0.004 |
| | ROUGE | – | – | 0.937 ± 0.008 |

of researchers. Each step is also finished in such a manner that one would be able to continue without starting at the first one to facilitate the research process. Throughout our work, we utilize open-source data and code and provide access to our resources.

## From raw patent data to organic chemistry patents

The data utilized in this study originate from patent records because of their availability and structured format. Among the various patent databases available, the United States Patent and Trademark Office (USPTO) and the European Patent Office (EPO) databases were chosen for this research. The database selection criteria included: the database size, accessibility in bulk and freedom of usage, and the absence of any costs associated with accessing, using the data or license issues. This ensures that other researchers can replicate our study using the same data without incurring additional costs. Other notable patent databases include the World Intellectual Property Organization (WIPO) Patent Collection, The Japan Patent Office (JPO) Database, The China National Intellectual Property Administration (CNIPA) Database, and The Korean Intellectual Property Office (KIPO) Database. However, some may have specific limitations or costs associated with accessing bulk data.

The US Patent and Trademark Office (USPTO) makes a database of patent applications and grants available *via* its bulk data storage system (https://bulkdata.uspto.gov, accessed on 8 of March 2022), which can be accessed free of charge. This database covers patents dating from 1971 and includes approximately 8.2 million patents with a total size of approximately 700 GB. Utilizing scripts or software to aid in the downloading process is recommended. The data is available for use, reuse, and distribution under the terms outlined in the USPTO's Open Data statement (https://developer.uspto.gov/about-open-data, accessed on 8 of March 2022).

The European Union (EU) Patent Database is hosted on the Google Cloud Platform (GCP). Information on downloading the database and a user manual can be found on the

European Patent Office's website (https://www.epo.org/searching-for-patents/data/bulk-data-sets/text-analytics.html, accessed on 8 of March 2022). While the data is available at no cost, there are charges associated with transferring it from GCP to local storage. The database includes a total of 3.7 million patents in English, with a total size of approximately 255 GB. Earliest patents in the database date back to 1978. The data is available under the Creative Commons Attribution 4.0 International Public License.

In order to utilize the patent data in a programming environment, it was first necessary to convert the data into a format that could be processed. While the European Patent Office (EPO) data set is in a uniform format and structure, the United States Patent and Trademark Office (USPTO) data set employs a variety of formats. Therefore, specialized scripts were written for each format to facilitate subsequent processing steps. The second major step involved extracting patents that pertained to a specific classification tag of organic chemistry. The USPTO data set includes classification identifiers, while the EPO data set does not. To overcome this, the EPO's freely available Application Programming Interface (API) was utilized to collect the classification data separately, resulting in a list of patent IDs with the organic chemistry classification tag. After this step, 230,594 EPO patents and 511,959 USPTO patents remained for further analysis. It is recommended for researchers to start with the dataset created after this step because it is significantly smaller, and the progress up to this step has been just the collection of data by straightforward programming algorithms.

## From organic chemistry patents to procedure paragraphs

We developed a methodology for classifying individual paragraphs within patterns to separate chemical synthesis procedures from other texts. This approach leverages deep neural networks and has been specifically designed by our research team for this purpose (referred to as Task 1). We successfully classified 182,315,235 paragraphs through this process, identifying 3,564,546 as relevant synthesis procedures after removing duplicate procedures. The paragraphs were additionally processed, removing unsuitable paragraphs that reference other procedures or do not contain full names of reactants or products.

Subsequently, we converted each paragraph into a more accessible format using a model developed during our research (Task 2). This new format facilitated more efficient processing, allowing easy access to distinct actions, action parameters, and chemical name extraction. Each paragraph was split into sentences using the *spaCy* library (*Honnibal & Montani, 2017*) and transformed the text into a structured format, which could then be reconstructed into a complete paragraph. As a result, our dataset comprised 3,058,295 unique synthesis procedures.

## From procedure paragraphs to the EPO/USPTO dataset

Structured paragraphs were used to extract the names of reactants and products in chemical synthesis processes. Reactants were identified through the ADD and MAKESOLUTION actions, while the YIELD action denoted the product. The extracted compound names were either in the IUPAC nomenclature or had trivial naming. These

actions were chosen because they represented the addition of reactants, while the action YIELD provided access to the products.

A multi-step process was developed to convert all compound names SMILES notation, which enables compound and substructure searching and is compatible with popular chemical programming libraries. Three translation methods were identified as helpful for this task based on their accuracy and coverage: (1) the PubChem application programming interface (API), which provides a robust platform for translating compound names in both trivial and IUPAC forms (*Kim et al., 2022*); (2) Open Parser for Systematic IUPAC Nomenclature (OPSIN), a versatile open-source name-to-structure conversion tool that excels in handling complex IUPAC names (*Lowe et al., 2011*); and (3) the *ChemAxon* application *Instant JChem*, a robust structure database management tool that supports various naming conventions (http://www.chemaxon.com, accessed on 5 December 2022).

The chemical entity detector *ChemDataExtractor 2.0* facilitated the process due to its compatibility with Python, extensive documentation, and GPU-accelerated usage (*Mavračić et al., 2021*). This tool scans text and returns chemical entities, specifically compound names. These features were crucial in efficiently processing large volumes of text while minimizing errors. Using *ChemDataExtractor 2.0*, a comprehensive list of all compounds was generated from procedure paragraphs in the dataset. Subsequently, a dictionary mapping IUPAC or trivial names to SMILES was created using the three translation methods. This approach increased the total number of translated names, as each method individually translated only about 60 to 70% of the input names. No known solutions exist for IUPAC to SMILES translation other than manual methods, which are infeasible for a large vocabulary of 3,709,454 items. The generated vocabulary was employed to convert the names of reactants and products to SMILES notation and are put into separate columns in the final dataset. Lastly, common solvents (*e.g.*, water or ethyl acetate) were removed from the reactant column, as they occasionally appeared in the *ADD* or *MakeSolution* steps and served only as the synthesis medium. This step ensured that only the reactants and products remained, rather than various solvents facilitating the synthesis and the work-up process. To enable further research and investigation, we've made our final dataset and the source code for our extraction and transformation scripts available to the public. The dataset, in a machine-readable format, and the Python scripts can be accessed from our project repository on GitHub (https://github.com/Mantas-it/ActionExtraction). Description and examples are provided within the repository to facilitate the understanding and utilization of both the dataset and the models.

The dataset encompasses a comprehensive range of organic compounds with varied structural characteristics. These include small organic molecules, organometallics, and biomolecules, each with distinct functional groups, including but not limited to, carbonyls, amines, alkenes, alkynes, and aromatic systems. Each of these functional groups can partake in various types of organic reactions such as addition, substitution, elimination, and rearrangement reactions, providing a broad coverage of synthetic pathways. These compounds participate in an assortment of organic reactions, reflecting a comprehensive representation of the reaction landscape in organic synthesis. From foundational reaction types such as substitution, addition (electrophilic, nucleophilic), and elimination, to more

advanced and specialized reactions such as pericyclic reactions, cross-coupling reactions and photochemical transformations, our dataset captures a spectrum of mechanistic possibilities.

## DISCUSSION

### Task 1: overall results

Table 6 presents the results for Task 1, where a BiLSTM classifier with a vocabulary size of 16,000 words achieved the highest accuracy of 0.939. This accuracy significantly surpasses the random (accuracy-0.553) and majority (accuracy-0.664) baselines, demonstrating the method's suitability for the solving task. Classifiers with a vocabulary size of 10,000 underperformed, likely due to a lack of expressiveness or the presence of out-of-vocabulary words. On the other hand, classifiers with vocabulary sizes larger than 16,000 did not yield better results due to data sparsity, as the probability of observing specific words decreases when the vocabulary size increases. It is worth noting that the dataset utilized for this task is considerably smaller than those commonly used in NLP tasks, which typically consist of millions of examples. The superiority of the BiLSTM model can be credited to its distinctive architecture that effectively utilizes the input in two directions, specifically from-past-to-future (forward) and from-future-to-past (backward) information. This bidirectional context assimilation is critical in making sense of complex chemical procedures where dependencies can exist in either direction. The application of LSTM within this architecture contributes significantly to the model's effectiveness. Firstly, LSTMs ensure more stable learning by controlling the flow of gradients throughout time, which alleviates the prevalent vanishing/exploding gradient problem. This feature is particularly important in capturing long-term dependencies across words and sentences, a common characteristic in scientific text. Secondly, LSTMs are designed to be more robust to noise or incomplete data. Their memory cells selectively retain, or discard information based on the determined importance of the input. This capability allows them to filter out noise and accurately capture the information conveyed in the data. Lastly, LSTMs' inherent capacity to retain or forget information as required, serves as a critical function in discerning the intricate details embedded in chemical procedures. The architectures of models can be found in the project's repository (https://github.com/Mantas-it/ActionExtraction).

A direct comparison of our results with previously reported findings is not feasible, as we employed a custom training dataset for this task. However, we also compared our approach with traditional ML methods such as naïve Bayes (accuracy-0.810, precision-0.753, recall-0.799, F1-score-0.768) and support vector machines (SVM) (accuracy-0.885, precision-0.837, recall-0.882, F1-score-0.855), both of which yielded considerably high results (especially SVM) but significantly worse compared to our best BiLSTM model. Although the optimal BiLSTM classifier's accuracy is relatively close to traditional ML methods, improving the scores further remains challenging due to the difficulty in distinguishing similar paragraphs. An essential metric to consider is recalled, as high scores in other metrics coupled with a low recall could indicate a substantial proportion of paragraphs being falsely identified as procedures. In this context, it is preferable to miss

actual procedures rather than have various paragraphs mixed in, as this would complicate downstream processing when preparing the dataset.

With their attention layers, transformer neural networks are capable of handling long-term dependencies, making them seemingly ideal for tasks involving paragraph inputs. However, as observed in Table 6, the performance of transformer networks ranks second best. Notably, during the training process, these networks exhibited rapid overfitting. One potential explanation for this phenomenon is that the training dataset might be too small for transformer neural networks to train effectively, given that they possess a substantially larger number of weights than the other classifiers tested in our research. Considering the superior performance of transformer networks in numerous other NLP tasks, it can be hypothesized that a larger dataset would enable transformers to outperform other classifiers. Besides, instead of training our transformer models, existing pre-trained ones (*e.g.*, BERT or sentence transformers) can be incorporated and later fine-tuned for our classification problems. This research direction is in our immediate plans. Despite it, there is always a risk that the available models may have too little chemistry knowledge.

## Task 1: error analysis and important observations

When analyzing the mislabeled instances, we found that the top-performing BiLSTM classifier can effectively differentiate between synthesis procedures and other paragraphs with high confidence. However, most errors occur when synthesis paragraphs deviate from the expected format. As outlined in section describing the data, paragraphs should contain full names of products and reactants without references to other procedures or compounds. The task's complexity increases when phrases such as *title compound* or the desired product are used with the product name. While the classifier can handle such cases to some extent, it is not consistently successful. Rare instances where *title compound* is followed by the full compound name—provided the remainder of the paragraph is reasonable—require nuanced classification. Increasing the dataset size with more instances could potentially enhance accuracy scores. In practical terms, the existing errors are somewhat mitigated during further processing of identified procedures when synthesis paragraphs containing specific key phrases are excluded from the resulting database.

The top BiLSTM classifier is approximately 3–5 times faster when used in an interface mode than the top transformer classifier, allowing it to classify the current dataset of 170 million paragraphs in about 12 h using a single GPU. Processing speed is crucial, as the next potential iteration of this task involves developing a similar methodology for open-source chemistry research articles to source more synthesis procedures, likely necessitating the processing of a far greater number of paragraphs than those available from patents.

## Task 2: overall results

Table 7 describes the results for Task 2, in which a transformer with a vocabulary size of 16,000 words attains the highest BLEU score of 0.951. As observed in the first task, a vocabulary of 16,000 words allows models to achieve the best performance. LSTM and BiLSTM models are less suitable for this task, as the optimal BiLSTM model only achieves a BLEU score of 0.671, significantly lower than the transformer model's score. Table 7 does

not include T5 small and base test scores with any other vocabulary sizes other than the *32,000 vocabulary* size column because a pre-trained T5 model and its tokenizer, which has a vocabulary of 32,128, were utilized. The fine-tuned T5 models produced BLEU scores comparable to the best model (T5 small–0.906, T5 base–0.902). Although the T5 model is considered more advanced for NLP tasks, its pre-trained status on a large *corpus* with limited relevance to the task at hand may not provide significant advantages. In contrast, transformer networks trained from scratch can efficiently learn to translate unstructured synthesis procedure text into a structured format. Transformers outperform other models mainly due to their unique architecture that enables the capture of dependencies regardless of their distance in the text, courtesy of the self-attention mechanism. This feature is particularly crucial in deciphering scientific language, where the context is often dependent on non-local interactions within the text. Moreover, the transformer model does not necessitate the sequential processing of data, which allows for better utilization of parallel computing resources, consequently enabling efficient model training. Also, the multi-head attention within transformers is able to focus on different parts of the input sequence while translating, allowing the model to handle varying contexts effectively, thus enhancing the quality of translation. These results are comparable to scores achieved by a transformer model (BLEU–0.850) (*Vaucher et al., 2020*) and a fine-tuned T5 small model trained for multiple chemistry-related tasks (BLEU–0.953) (*Liu et al., 2019*). It is important to note that our study employed a distinct dataset specifically created for this task, which includes additional parameters for one step, as described in the data section. The achieved high BLUE value encourages us to experiment with this problem even more. Input texts can be modified (paraphrasing, summarizing, back-translating, or by hand) by making them even less formalized and, therefore, better comprehensible for non-chemistry specialists. On the other hand, the output can become even more formalized and understandable to robots. However, the existence of such a method could bring chemistry to a broader circle of people (especially in the education sector). Thus, the gap between the level of formalization between the input and the output would become much larger than it is right now, and therefore the solving problem would become even more challenging. Nevertheless, it is an exciting future research direction.

## Task 2: error analysis and important observations

The optimal model for this task attains a BLEU score of 0.953 and generally makes very few errors. Accuracy has been evaluated by measuring how many sentences are predicted identically, as they are the testing dataset, and it achieved a score of 0.823. The most common mistakes involve minor details, such as the order of additional actions when multiple components are added successively. In some cases, the model fails to detect the amount of the added compound and does not include it in the structured sentence format. Errors mainly occur with the actions *InvalidAction* and *FollowOtherProcedure*, which are highly specific and infrequent. It was possibly not provided with enough examples for the model to learn each case effectively. *InvalidActions* refers to unfeasible actions, such as when a sentence describes a reaction mechanism, while *FollowOtherProcedure* denotes a step where another procedure or method is referenced in a patent. Moreover, certain

sentences only partially describe a step and then reference a similar step in other procedures, making translation challenging. The distinction between these two actions can be subtle and difficult for an expert to discern during manual dataset labeling. In large-scale data processing, it is advisable to exclude paragraphs with *InvalidAction* and *FollowOtherProcedure* actions, as these include either incorrect operations or content that surpasses the model's comprehension, thereby undermining the ability to generate sensible translations. In the range of available procedure sources, these specific actions are encountered in approximately 3 to 7% of all instances. Conversely, in an experimental laboratory context where procedure processing is conducted on a smaller scale and manual verification of translations is feasible, it becomes beneficial to conduct thorough examinations of these instances. This is because not all occurrences of *InvalidAction* and *FollowOtherProcedure* are necessarily detrimental to the structure of the converted procedure. Most crucially, no instances of incorrectly converted compound titles have been observed.

The fine-tuned T5 small model attains a lower BLEU score of 0.906. The tokenizer used with the pre-trained T5 model can be fine-tuned by training it on a task-specific *corpus* and adjusting the vocabulary to include unique characters and words that might appear. Although such tests were conducted, the model achieved even lower scores (BLEU-0.780). This outcome may be attributed to the tokenizer significantly altering the vocabulary when trained on synthesis procedures, with only around 2,000 tokens out of 30,128 remaining the same. This drastic change likely necessitates training the original embedding weights alongside the rest of the model. Consequently, the model does not benefit from pre-training and would need to be trained from scratch on a large chemistry text *corpus*, which is beyond the scope of this article.

The fine-tuned T5 model can accurately translate unstructured procedure sentences with minimal errors. Most of the mistakes resemble those previously described, occasionally including incorrect quantities, such as copying the original expression from the sentence as (about 98 mg) instead of (98 mg). However, it is worth noting that the model can still be employed for applications in contexts where procedures are relatively simple and devoid of complex or misleading information or expressions. Furthermore, the T5 model is used with the Python library *HuggingFace*, which offers convenient tools for setup, interface, and integration compared to transformer models trained with *OpenNMT*, which require several steps for setup. However, the T5 model is significantly larger, and the translation process is slower than the transformer models.

## CONCLUSIONS AND FUTURE WORK

In conclusion, the article demonstrated the effectiveness of BiLSTM and Transformer models in tackling the challenging tasks of identifying synthesis procedures (as a text classification problem) and translating unstructured synthesis text into a structured format (as a text generation problem). The BiLSTM classifier achieved the best classification accuracy, equal to 0.939, outperforming traditional ML approaches and (random and majority) baselines, proving its suitability and reliability. The Transformer model attained the highest BLEU score, equal to 0.951 in the unstructured to structured text translation

problem. We also provide the methodology, which utilizes the optimal neural network models analyzed in the study, to construct a dataset comprising synthesis procedures, their structured counterparts, reactants, and products. Moreover, we make this research's models, methods, and datasets publicly available.

Future work should focus on expanding the training dataset for both tasks because more training data could potentially enhance the performance of Transformer models for our classification problem and reduce errors associated with rare actions for our translation problem. Furthermore, developing methods for open-source chemistry research articles could enable sourcing more synthesis procedures, while continued research and refinement of these models will contribute to a deeper understanding of their capabilities and limitations in chemistry-related tasks.

### Funding
This work was supported by the Vytautas Magnus University and JSC Synhet. The funders had no role in study design, data collection and analysis, decision to publish, or preparation of the manuscript.

### Grant Disclosures
The following grant information was disclosed by the authors:
Vytautas Magnus University.
JSC Synhet.

### Competing Interests
Mantas Vaškevičius and Liudas Šlepikas are employed by JSC Synhet.

### Author Contributions
- Mantas Vaškevičius conceived and designed the experiments, performed the experiments, analyzed the data, performed the computation work, prepared figures and/or tables, authored or reviewed drafts of the article, and approved the final draft.
- Jurgita Kapočiūtė-Dzikienė analyzed the data, authored or reviewed drafts of the article, and approved the final draft.
- Arnas Vaškevičius analyzed the data, performed the computation work, prepared figures and/or tables, and approved the final draft.
- Liudas Šlepikas analyzed the data, authored or reviewed drafts of the article, and approved the final draft.

### Patent Disclosures
The following patent dependencies were disclosed by the authors:
The US Patent and Trademark Office (USPTO) makes a database of patent applications and grants available *via* its bulk data storage system (https://bulkdata.uspto.gov, accessed on 8 of March 2022), which can be accessed free of charge. This database covers patents dating from 1971 and includes approximately 8.2 million patents with a total size of

approximately 700 GB. Utilizing scripts or software to aid in the downloading process is recommended. The data is available for use, reuse, and distribution under the terms outlined in the USPTO's Open Data statement (https://developer.uspto.gov/about-open-data, accessed on 8 of March 2022).

The European Union (EU) Patent Database is hosted on the Google Cloud Platform (GCP). Information on downloading the database and a user manual can be found on the European Patent Office's website (https://www.epo.org/searching-for-patents/data/bulk-data-sets/text-analytics.html, accessed on 8 of March 2022). While the data is available at no cost, there are charges associated with transferring it from GCP to local storage. The database includes a total of 3.7 million patents in English, with a total size of approximately 255 GB. Earliest patents in the database date back to 1978. The data is available under the Creative Commons Attribution 4.0 International Public License.

## Data Availability

The raw patent text data was obtained from two sources:

(1) USPTO bulk data storage (https://bulkdata.uspto.gov/)

(2) EPO text data for analytics (https://www.epo.org/searching-for-patents/data/bulk-data-sets/text-analytics.html).

All secondary datasets created in our research, methods, and code used to process the data and train models are available at GitHub: https://github.com/Mantas-it/ActionExtraction.

Code is also available at Zenodo:

Mantas-it. (2023). Mantas-it/ActionExtraction: First_release (v1.0.0). Zenodo. https://doi.org/10.5281/zenodo.8158599.

## Supplemental Information

Supplemental information for this article can be found online at http://dx.doi.org/10.7717/peerj-cs.1511#supplemental-information.

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
