# Peer review of "Deep learning-based automatic action extraction from structured chemical synthesis procedures"

_PeerJ Computer Science, doi:10.7717/peerj-cs.1511_

## Round 0.1 · original submission · Minor Revisions

The reviewers have minor concerns about this manuscript. The authors should provide point-to-point responses to address all the concerns and provide a revised manuscript with the revised parts being marked in different color.

·

Basic reporting

no comment

Experimental design

no comment

Validity of the findings

no comment

Additional comments

Chemical informatics has become an indispensable aspect of modern chemical research. This paper uses artificial neural networks, such as short-term memory, bidirectional LSTM, converter, and fine-tuning T5, to extract relevant data from USPTO and EPO patents, and designs a pipeline that can use original patent data to create chemical reactions and their processes in a structured format, to modify or merge new patents easily. A method of using a machine learning algorithm to extract actions from a structured chemical synthesis program was proposed, which bridged the gap between chemistry and natural language processing. This study promotes the application of AI-based methods, which simplify synthesis pathways, predict reaction results, and optimize experimental conditions.
The following modifications are suggested:
1. Hope to further clarify the chemical characteristics of the data set collected in the article.
2. The article should establish a link to facilitate the reader to refer to the dataset and program.

Reviewer 2 ·

Basic reporting

The paper is very well-written, well-structured and provides detailed descriptions of the proposed methodology and its evaluation. There is only a minor expression issue

Line 58~59 "which can identify patterns and trends in data that humans may not discern easily discernible" I believe there are duplicate usage of discernible

Experimental design

The paper proposes a methodology that uses machine learning algorithms to extract actions from structured chemical synthesis procedures, bridging the gap between chemistry and natural language processing. The methodology includes a pipeline that combines machine learning algorithms and scripts to extract relevant data from USPTO and EPO patents by collecting, processing, and transforming experimental procedures into a series of structured actions. The pipeline has two primary tasks solved by ML algorithms: classification of patent paragraphs to select relevant information and extraction of actions from the selected paragraphs. The paper provides detailed descriptions of each step in the pipeline and evaluates the performance of the methodology on a dataset of chemical synthesis procedures.

Great work!

Validity of the findings

The evaluation results show that the proposed methodology achieves high precision and recall scores, indicating that it is effective in extracting relevant information from patent documents, which is quite convincing.

Reviewer 3 ·

Basic reporting

The paper organizes well but is too long to read, the author may reorganize the section to make it better readable.

Experimental design

The experimental design and data are enough to explain the objective of the paper.

Validity of the findings

The innovativeness is moderate.

Additional comments

1. The result shows that BiLSTM performs the best in task 1, could the author give more distinct explanations about why this ML model achieved the highest score?
2. Same for task 2, could the author provide more detail about why transformers give the best results compared with other models?
3. Could the author explain how we deal with the sentence with InvalidAction and FollowOtherProcedure?

---

## Round 0.2 · accepted · Accept

All reviewers are satisfied with the current revised manuscript. I concur with reviewers and suggest accepting this manuscript.

·

Basic reporting

no comment

Experimental design

no comment

Validity of the findings

no comment

Additional comments

This manuscript has been completely revised. This paper will inspire and help many colleagues.

Reviewer 2 ·

Basic reporting

N/A

Experimental design

N/A

Validity of the findings

N/A

Additional comments

The revision resolved all my questions and concerns. No additional comments.

Reviewer 3 ·

Basic reporting

The paper is well organized.

Experimental design

The data is sufficient.

Validity of the findings

After reviewing the revised version, I recommend accepting this paper.

Additional comments

After reviewing the revised version, I recommend accepting this paper.